# Revisiting the Phylogenetic Relationship and Evolution of Gargarini with Mitochondrial Genome (Hemiptera: Membracidae: Centrotinae)

**DOI:** 10.3390/ijms24010694

**Published:** 2022-12-30

**Authors:** Feng-E Li, Lin Yang, Jian-Kun Long, Zhi-Min Chang, Xiang-Sheng Chen

**Affiliations:** 1Institute of Entomology, Guizhou University, Guiyang 550025, China; 2The Provincial Special Key Laboratory for Development and Utilization of Insect Resources, Guizhou University, Guiyang 550025, China; 3The Provincial Key Laboratory for Agricultural Pest Management of Mountainous Regions, Guizhou University, Guiyang 550025, China

**Keywords:** Auchenorrhyncha, treehopper, Gargarini, mitogenome, phylogenetics

## Abstract

**Summary:**

This study attempts to elucidate the taxonomic complexities in tribe Gargarini using phylogenetic and evolutionary analysis based on the integrated morphological and molecular datasets: the shape of the pronotum and the complete mitochondrial genome, respectively. Despite being the largest tribe of the subfamily Centrotinae, with more than 400 species, the morphological similarities among the genera and complicated taxonomic history, the phylogenetic relationships among many species are still unknown. The phylogenetic relationships among some species of this tribe based on the mitochondrial genome were established in this study. Moreover, this study provides the possibility of establishing an evolutionary history. This study provides a basis for follow-up research on the phylogeny and evolution of more species of this tribe.

**Abstract:**

In this study, we newly sequenced and analyzed the complete mitochondrial genomes of five genera and six species in Gargarini: *Antialcidas floripennae*, *Centrotoscelus davidi*, *Kotogargara minuta*, *Machaerotypus stigmosus*, *Tricentrus fulgidus*, and *Tricentrus gammamaculatus*. The mitochondrial genomes contain 13 protein-coding genes, two ribosomal RNA genes, 22 transfer RNA genes, and a control region. The lengths of the mitochondrial genomes are 15,253 bp to 15,812 bp, and the AT contents of the obtained mitogenomes indicate a strong AT bias, ranging from 75.8% to 78.5%. The start codons of all PCGs show that most start with a typical ATN (ATA/T/G/C) codon and less start with T/GTG; the stop codon TAA is frequently used, and TAG and a single T are less used. In Gargarini mitogenomes, all tRNA genes can be folded into the canonical cloverleaf secondary structure, except for *trnaS1*, which lacks a stable dihydrouridine (DHU) stem and is replaced by a simple loop. At the same time, the phylogenetic analysis of the tribe Gargarini based on sequence data of 13 PCGs from 18 treehopper species and four outgroups revealed that the 10 Gargarini species form a steady group with strong support and form a sister group with Leptocentrini, Hypsauchenini, Centrotini, and Leptobelini. Diversification within Gargarini is distinguished by a Later Cretaceous divergence that led to the rapid diversification of the species. Moreover, the ancestral state reconstructions analysis showed the absence of the suprahumeral horn, which was confirmed as the ancestor characteristic of the treehopper, which has evolved from simple to complex. Our results shed new light specifically on the molecular and phylogenetic evolution of the pronotum in Gargarini.

## 1. Introduction

The treehopper family Membracidae (Hemiptera: Auchenorrhyncha) is renowned for their exaggerated pronotal “helmet” (Figure 1) and comprises approximately 3200 described species [1]. This large group of phytophagous insects also includes some pests of horticulture crops and forestry because they feed on the phloem of many different plant species [2]. The tribe Gargarini Distant, 1908, is one of the largest tribes of the subfamily Centrotinae, with more than 400 described species. It has been placed as the sister group of Boocerini [3], Terentiini [4], or Antialcidini [2] in recent phylogenetic studies based on different types of datasets ranging from morphological to genomic levels. Morphologically, species of Gargarini exhibit different pronotal structures, such as suprahumeral horns showing a bracket ridge or not (genera *Machaerotypus*, *Gargara*, and *Kotogargara*). In addition, they have a flat or raised posterior pronotal process (genera *Nondenticentrus*, *Pantaleon*, etc.). These variations in pronotal structures make this tribe controversial [2,3,4]. Previous studies have explored this issue and have focused primarily on the morphology-based taxonomy of a broad group or other tribes [2,5,6,7]. So far, Gargarini still has different definitions. Despite extensive previous studies of the tribe Gargarini [2,3,4,5,6,7,8,9,10], several chief questions on the evolution of Gargarini remain unresolved. One regards the largely undetermined phylogenetic relationships at generic and species levels within the tribe. Because of this, little is known about the timescale of their divergence and the morphological evolution of Gargarini. Furthermore, the differential diagnosis of the tribe Gargarini is another key issue. 

The diagnostic features of the pronotum of the genus *Gargara* Amyot and Serville, 1843, which belongs to Gargarini, are only defined in terms of humeral angle and posterior pronotal process. However, there are still a small number of species without suprahumeral horns placed in *Tricentrus* Stål, 1866, not *Gargara* Amyot and Serville, 1843. The genus *Tricentrus* is grouped with two other genera into the tribe Tricentini [2]. It was established by Ahmad and Yasmeen in 1974 and attached with genus diagnosis: the metathoracic trochanter with spines; hindwing vein R branched into R_1_, R_2+3_, and R_4+5_, vein M branched into M_1+2_ [7]. The validity of this tribe is recognized by Yuan and Chou in later studies [2]. What is more, one study about phylogenetic analysis on the subfamily Centrotinae provides new evidence to help understand the intergeneric phylogeny of Gargarini yielded thatTricentini is considered a new synonym for Gargarini [3]. In general, the results of the phylogeny of Gargarini are inconsistent.

On the other hand, molecular markers are gradually applied to the phylogenetic analysis of insects, including treehoppers, with the development of molecular techniques [8,9,10]. While some molecular data were added in subsequent studies, only a few species of Gargarini were involved in these analyses [11,12,13,14]. Therefore, adding other additional sources, such as complete mitogenomes, may help to clarify the composition of this tribe and promote phylogenetic resolution with other tribes. Knowledge of phylogenetic relationships among members of the tribe Gargarini may help to elucidate their evolutionary history and systematic position of different taxa at generic and species levels. Moreover, elucidating mitochondrial genomes (mitogenomes) may offer a better understanding of phylogenetic relationships among the members of Centrotinae. Currently, 15 complete or partial mitogenomes sequences of treehoppers have been deposited on GenBank as of February 2022 [10,14,15,16,17,18,19], of which four species, *Gargara genistae*, *Maurya qinlingensis*, *Tricentrus brunneus*, and *Tricentrus* sp., belong to Gargarini [13,15].

In this study, the following mitogenomes were sequenced and annotated for six Gargarini species: *Antialcidas floripennae*, *Centrotoscelus davidi*, *Kotogargara minuta*, *Machaerotypus stigmosus*, *Tricentrus fulgidus*, and *Tricentrus gammamaculatus*. Their structures and basic characteristics, as well as comparative mitogenomics of these sequences, will are provided. The phylogenetic analyses were performed using maximum likelihood (ML) and Bayesian inference (BI) methods based on available mitogenomes in Gargarini. In addition to this, we estimate the timescale of cladogenesis within Gargarini based on molecular dating methods and trace the morphology of the pronotum history of this tribe by ancestral state reconstruction.

## 2. Results

### 2.1. Genome Organization and Composition

The complete mitogenomes of six Gargarini species: *Centrotoscelus davidi* (GenBank ID: OP714179; length: 15,618 bp), *Kotogargara minuta* (GenBank ID: OP714178; length: 15,253 bp), *Tricentrus fulgidus* (GenBank ID: OP714176; length: 15,479 bp), *Tricentrus gammamaculatus* (GenBank ID: OP714177; length: 15,812 bp), *Antialcidas floripennae* (GenBank ID: NC_065325; length: 15,612 bp), *Machaerotypus stigmosus* (GenBank ID: OP714181; length: 15,382 bp) (Figure 2), and one Centrotini treehopper: *Centrotus cornutus* (secondary sequencing, Appendix A) (GenBank ID: OP714180; length: 15,090 bp) were identified as circular double-stranded molecules. The size mitogenomes longest and shortest are *T. gammamaculatus* and *C. cornutus,* respectively, and updated previous data [13,14,15,16,17,18,19]. The 37 typical mitochondrial genes (13 PCGs, 22 tRNAs, and 2 rRNAs) and the control region are contained in mitogenomes (Appendix A). They showed identical gene orders with the typical gene arrangement of insects [20,21]. The major strand (J-strand) encoded 23 genes (9 PCGs and 14 tRNAs), while the remaining 14 genes (4 PCGs, 8 tRNAs, and 2 rRNAs) are transcribed on the minor strand (N-strand) (Appendix A). All Gargarini species shared four conserver overlap regions in *trnI*-*trnQ* (3 bp: TTG), *trnY-cox1* (2 bp: AT), *atp8*-*atp6* except *T. fulgidus* (7 bp: ATGATAA), *nd4*-*nd4L* (7 bp: TTATCAT). These overlaps are also found in previous studies of treehoppers [16,17,18,19].

Detailed information on nucleotide composition and skewness levels are shown in Appendix A. Among, AT content of the mitogenomes indicated a strong AT bias, as in other published Membracidae insects [13,14,15,16,17,18,19], ranging from 75.8% in *A. floripennae* to 78.7% in *T. brunneus*. The control region had the highest AT content (72.7–89.4%) except *C. davidi*, *Teicentrus* sp., and *M. qinlingensis*, whereas rRNAs had the lowest AT content (71.4.6–71.9%) except *A. floripennae*, *G. genistae*, *C. davidi*, *T. gammamaculatus*, *K. minuta*, *M. qinlingensis*, and *M. stigmosus*. The results of complete mitogenomes of Gargarini exhibited positive AT skew (0.1140–0.1990) and negative GC skew (−0.2320 to −0.1200), corresponding with the AT bias observed in other insects [17,22,23,24,25].

### 2.2. PCGs and Codon Usage

The 13 PCGs lie in all seven treehopper mitogenomes, with this length, and proportion of AT content ranging from 10,862 bp to 10,912 bp and 73.6% to 77%, respectively. Meanwhile, *atp8* and *nad5* were the shortest and longest separately in Gargarini mitogenomes. The AT (−0.1528 to −0.1163) skewness values of PCGs were similar between different genera of tribe Gargarini treehoppers, yet GC (−0.05 to 0.012) skewness values were different. All 13 PCGs show a negative AT skew. The start and stop codons of all PCGs are listed in Appendix A, showing that most started with typical ATN (ATA/T/G/C) codons, except for *nad5* and *nad1* of *K. minuta*, *T. gammamaculatus*, *A. floripennae*, and *M. stigmosus*, *nad5* of *C. davidi*, *nad1* of *T. fulgidus*, which started with TTG. The codon TAA is frequently used as the stop codon, and TAG or a single T are less used.

The number of codon usage and relative synonymous codon usage (RSCU) of PCGs in present Gargarini mitogenomes were calculated and demonstrated. The results show that the four most frequently used codons are UUA (L), UCA/U (S), and CGA (R), yet CUC/G (L), AGC (S), GGC (G) are rarely used (Figure 3 and Appendix A). Similar results appear in other Membracidae or leafhopper mitogenomes [16,17,18,19,22].

The nonsynonymous substitutions (Ka), synonymous substitutions (Ks), and Ka/Ks (ω) values are estimated for each PCG of the 18 treehoppers of subfamily Centrotinae, and the details shown in Figure 4 and Appendix A. The Ka/Ks values of all PCGs were less than 1 and ranged from 0.132 for *cox1* to 0.763 for *atp8*, indicating that the 13 PCGs were under purifying selection. The evolution rate of 13 PCGs was *atp8* > *nd6* > *nd2* > *nd5* > *nd4* > *nd4l* > *nd3* > *atp6* > *nd1* > *cox3* > *cox2* > *cytb* > *cox1* successively. Among *cox1* showed the strongest purifying selection, and *atp8* showed the least selection pressure and fastest evolution speed among the protein-coding genes of Centrotinae.

### 2.3. Transfer and Ribosomal RNA Genes

The large and small ribosomal RNA genes are encoded by the minor strand and located between *trnL1* to *trnV* and *trnV* to the control region, respectively. The total length of rRNAs among 11 sequenced mitogenomes ranges from 1896 to 2014 bp, with the AT contents ranging from 71.4 to 81.7%, positive GC skew, and negative AT skew.

All 22 tRNA genes of 11 treehopper mitogenomes, including 6 newly sequenced, are determined, with a total length ranging from 1392 to 1426 bp, and the length of each tRNA is 59–72 bp. In the 22 tRNA genes, 14 were encoded by the major strand, and the remaining 8 were encoded by the minor strand, with positive AT skew except for *M. stigmosus* and *C. cornutus*, and positive GC skew. All tRNA genes in the seven new mitogenomes could be folded into the canonical cloverleaf secondary structure except for *trnaS1* lacked a stable dihydrouridine (DHU) stem, which is replaced by a simple loop. A total of six mismatched base pairs (G-U, C-U, U-U, A-C, A-A, A-G, as well as extra A/U/G nucleotide) were found in seven species predicted secondary structures (Appendix A).

### 2.4. Control Region

The controlling elements for replication and transcription were included in the control region with the longest non-coding region. It was located between *s-rRNA* to *trnaI* in seven treehopper mitogenomes and variable in length, ranging from 869 bp (*C. cornutus*) to 2261 bp (*T. brunneus*), with AT contents ranging from 72.7% (*Tricentrus* sp.) to 89.4% (*G. genistae*). The AT and GC skew were all positive, except for AT skew (−0.0132) of *G. genistae* (with a gap). The control region with the structural organization of seven treehoppers mitogenomes is illustrated in Appendix A. Two repeat regions were present in *C. davidi*, *T. gammamaculatus*, *T. fulgidus*, and one in the remaining four species, *K. minuta*, *A. floripennae*, *M. stigmosus*, and *C. cornutus*.

### 2.5. Phylogenetic Relationship

Phylogenetic analyses among 22 species of treehoppers (18 ingroup and 4 outgroup) were reconstructed based on nucleotide and amino acid sequence data of 13 PCGs using ML and BI analyses. The results yielded largely congruent topologies, with most branches receiving strong support (Figure 5, Appendix A). In phylogenetic trees, the subfamily Aetalioninae and Smillinae formed a steady group located at the basal of these trees. This finding is consistent with previous studies [13]. The remaining species belong to Centrotinae, located at the end of trees. In the subfamily Centrotinae, the four tribes (Leptocentrini, Hypsauchenini, Centrotini, and Leptobelini) gathered to form a sister group to Gargarini. The species of tribe Gargarini always clustered with high support (BP = 100; SH-aLRT = 100; PP = 1). Moreover, the result unsupported the tribe Centrotini with the sister-group relationship to Leptocentrini, Hypsauchenini, and Leptobelini. It is slightly different from some previous studies [13]. This result could be because we updated the mitochondrial genome data for the species *C. cornutus*. 

### 2.6. Divergence Time Estimates

The divergence time analysis using the differentiation time results postulated by Dietrich et al. [4] yielded dispersion times for most nodes that were consistent with the previous study (Figure 6 and Appendix A). The early divergence within Membracidae, referring to the split between subfamily Aetalioninae + Smillinae and the remaining species, was dated during the Early Cretaceous at 138.27 Ma (95% HPD = 134.05−148.06). The subfamily Smillinae diverged from Aetalioninae 129.51 Ma (95% HPD = 112.51−137.11). The divergence of the Centrotinae clade took place in 96.94 Ma (80.28−113.24). The tribes Leptocentrini, Leptobelini, Hypsauchenini, and Centrotini clade divergence emerged in the 84.27 Ma (95% HPD = 68.19−99.30). The remaining clade Gargarini divergence happen to Late Cretaceous at 77.58 Ma (95% HPD = 63.00−91.92).

### 2.7. Ancestral State Reconstructions

The ancestral state reconstruction analysis of the suprahumeral horn of the tribe Gargarini using Bayesian Binary MCMC analysis by RASP showed that the ancestral treehopper was absent the suprahumeral horn (Figure 6), and through eight times changes of character. In the tribe Gargarini, four types of changes of character were found. The three times transitions from suprahumeral horn absent to present appear in the Ino6, Ino8, and Ino13 branches, respectively. On the contrary, twice transitions from suprahumeral horn present to absent show up in branches Ino15 and Ino17 separately. The other two types of changes were from suprahumeral horn present to pronotum other situations and contrary situations. The former occurs once at branch Ino7 and Ino9, respectively, and pronotum with other situations to suprahumeral horn present occurs in branch Ino11 only one time.

## 3. Discussion

### 3.1. Genome Character and Phylogenetic Relationships

In this study, six treehopper mitogenomes from the tribe Gargarini were newly sequenced and analyzed. They showed identical gene orders with the typical gene arrangement of insects [20,21]. Among the six mitogenomes, the length ranged from 15,253 bp in *K. minuta* to 15,812 bp in *T. gammamaculatus*. The difference in length is mainly attributed to the difference in intergenic spacer regions and the length of the control region. The length of new and known sequenced Gargarini with control region from 1062 bp to 2261 bp (*Tricentrus brunaeus*). The length differences of each PCG in the six newly sequenced mitogenomes did not exceed 18 bp; this result suggests that the PCGs have relatively conserved characteristics among species. The length change of this tRNA and rRNA in different species was limited due to the stability of their secondary structures. Among all 22 tRNA genes, the *trnaS1* of six species tribe Gargarini lacked a stable dihydrouridine (DHU) stem and was replaced by a simple loop, which has been recognized in many metazoans [12,13,14,15,16,17,18,19]. The TAA codon was used as the stop codon for most PCGs in the tribe Gargarini, and TAG or a single T was less used. It is present common in insects [9,21]. Among the protein-coding genes of Centrotinae, *cox1* showed the strongest purifying selection, and *atp8* showed the least selection pressure and fastest evolution speed. The result was consistent with the studies of Cicadellidae or other insects [12,23,26,27]. In addition, the overlap 7 bp region between *atp8* and *atp6* has appeared in other insects, and the species of tribe Gargarini also show this characteristic. However, in the whole tribe, we knew this region was ATGATAA, while ATACTAA in *Tricentrus fulgidus*. It seems that is the difference between different species. However, the overlap of other species of the same genus was not different, which seems to be a key transition node in combination with phylogenetic analysis. The species, *Tricentrus fulgidus*, is located near the *Kotogargara minuta* in the branch of tribe Gargarini of the phylogenetic tree and formed sister groups with the remaining species. Morphologically, the head and pronotum of the species were smooth and hairless, which distinguishes it from other species.

On the other hand, the tribe Gargarini species formed a steady group with a high support value ((BP = 100; SH-aLRT = 100; PP = 1) in the phylogenetic analysis. This is a new finding on the phylogeny of Gargarini based on the complete mitochondrial genome data compared with the previous study with fewer participating species [11,12,13]. Moreover, this result is consistent with the conclusion based on the morphological analysis of Wallace’s paper [6]. In contrast, it differs from the results of Ahmad and Yasmeen (1974) and Yuan and Chou et al. (2002), who considered *Tircentrus* as a separate tribe. This contradictory result may be due to the fact that the tribe Gargarini includes the genus *Tricentrus* with nearly 300 species. In addition to the typical types with or without suprahumeral horn, there are a few special types of pronotum, for example, *Pantaleon*, and *Mesocentrina*. Nevertheless, our molecular results support the validity of this tribe. This result may be related to the small number of samples selected by the genus *Tricentrus* for analysis. In the future, we will focus on relevant research on this genus.

### 3.2. Character Evolution

In ancestral state reconstruction analysis, the morphology and evolution process of the suprahumeral horn of the ancestral treehoppers were preliminarily revealed. The suprahumeral horn absence is confirmed as the ancestor characteristic of treehopper and has evolved from simple to complex. It seems easy to understand. More and more molecular phylogenetic studies have largely supported the hypothesis that treehoppers are derived from leafhoppers [4,28,29,30]. Leafhoppers and treehoppers share some characteristics, such as male and female genitalia. Even primitive treehoppers like the subfamily Aetalioninae are similar to *Dryodurgades* in Megophthlminae in morphological characteristics. Additionally, this may be related to the same origin of the pronotum and wings of treehoppers [31,32]. The plasticity of this ancient mechanism provides the possibility for these diverse morphologies of pronotum.

Moreover, the influence of the environment is essential. This extreme morphological modification of treehoppers may play a role in crypsis, predatory defense, maternal care, and symbiosis [33]. The divergence of treehoppers began in the Early Cretaceous. Although the environment at this stage is suitable for the development of insects, a mass extinction event occurred in the Late Cretaceous for the planet hit the Earth [34]. At this stage, the Earth worsens in climate gradually after suffering frequent volcanic activity and large-scale regression. This may be the reason why the modern subfamily of treehopper lineage appeared in the late Cretaceous or early Paleogene [4]. The subsequent development of angiosperms may also promote the evolution of the treehopper lineage. According to the statistics of Yuan and Zhou, the host plants of treehoppers are almost angiosperms [2]. 

## 4. Materials and Methods

### 4.1. Sample Preparation

A total of seven species were included in the analyses: Gargarini with six species and Centrotini with one species. These included ingroup taxa used in this study were collected from China, and the detailed collection information is presented in Appendix A. All fresh specimens were immediately preserved in absolute ethanol and then stored at −20 °C at the Institute of Entomology, Guizhou University, Guiyang, China. These species were identified based on their morphological characteristics based on the following available source of literature [1,2,3,35,36].

### 4.2. Genomic DNA Extraction, Sequencing

Whole mitogenomes were sequenced using a next-generation sequencing platform with Illumina Hiseq 2500 at OriGene (Beijing, China). Following quantification of the total DNA that was extracted, an Illumina TruSeq library with single species was generated from the pooled genomic DNA, with an average insert size of 480 bp. This library was sequenced on a full run of Illumina Hiseq 2500 with 500 cycles and paired-end sequencing (250 bp reads).

### 4.3. Data Assembly, Annotation, Analysis

The sequencing datasets were assembled using MitoZ 2.4-alpha (Berlin, Germany) [37] under default settings and the mitogenome of *Leptobelus gazella* (Membracidae: Centrotinae; NC_023219) as a reference. The Geneious Prime 2021.1.1 (Auckland, New Zealand) [38] was used to annotate previously assembled mitogenomes. A total of 13 PCGs were predicted by determining their open reading frames using the invertebrate mitochondrial genetic codon; rRNA genes and AT-rich regions were identified by alignment with homologous genes from other treehopper species; the locations and secondary structures of tRNA genes were predicted using the tRNAscan-SE search server (http://lowelab.ucsc.edu/tRNAscan-SE), accessed on 25 August 2022 [39] and the MITOS web server (http://mitos.bioinf.uni-leipzig.de/index.py), accessed on 2 September 2022 [40] with invertebrate codon predictors. Mitogenomic circular maps were depicted using the OGDRAW—Draw Organelle Genome Maps (https://chlorobox.mpimp-golm.mpg.de/OGDraw.html), accessed on 15 September 2022 [41]. Tandem repeats in the AT-rich region were searched using the Tandem Repeats Finder program (http://tandem.bu.edu/trf/trf.basic.submit.html), accessed on 27 September 2022 [42].

Strand asymmetry was calculated according to the following formulas [43]: AT skew = (A − T)/(A + T) and GC skew = (G − C)/(G + C). A heatmap of the relative synonymous codon usage (RSCU) values of PCGs (excluding stop codons) in the Gargarini and Centrotini mitogenomes was drawn using TBtools v1.09832 (Beijing, China) [44]. The rates of nonsynonymous substitutions (Ka) and synonymous substitutions (Ks) of 13 aligned PCGs were determined by DnaSP v6.12.03 (Barcelona, Spain) [45]. These seven newly sequenced mitogenomes were submitted to GenBank with accession numbers OP714176, OP714177, OP714178, OP714179, OP714180, and OP714181.

### 4.4. Phylogenetic Analyses

In addition to the seven mitogenomes obtained in this study, the complete mitogenomes of 15 species were downloaded from GenBank for phylogenetic analyses. The four mitogenomes from Cercopoidea and Cicadoidea were used as outgroups, respectively. Detailed information and accession numbers of these mitogenomes are listed in Appendix A [14,15,16,17,18,19,20,46,47]. The nucleotide sequences of 13 PCGs were extracted using the introduction PhyloSuite v.1.2.2 (Beijing, China) [48]. The amino acid sequences were deduced to make use of MEGA v.7.0.26 (Pennsylvania, USA) [49] according to the mitochondrial codon of invertebrates. Individual gene alignments were concatenated into a database using SequenceMatrix v.1.7 [50]. The final dataset was analyzed using maximum likelihood (ML) and Bayesian inference (BI) inferred, respectively, using IQ-TREE v.1.6.8 (Wien, Austria) [51] and MrBayes v3.2.6 (Stockholm, Sweden) [52]. The best partition schemes and substitution models for ML and BI analyses of the dataset (Appendix A) were estimated using PartitionFinder v.2.1.1 (Sydney, Australia) [53]. The Edge-linked partition model for 10,000 ultrafast [54] bootstraps, as well as 1000 SH-aLRT [55] replicates, were set in ML analyses. BI analysis was performed under the following conditions: two independent runs each for 3,000,000 generations, sampling every 1000th generation, and the first 25% discarded as burn-in. When the average standard deviation of split frequencies fell below 0.01 and remained stable, stationarity was considered to have reached.

Furthermore, an effective sample size (ESS) value of more than 200 was used as a convergence diagnostic using TRACER v.1.7. (Edinburgh, UK) [56]. When the bootstrap percentage (BP) was >75% or the Bayesian posterior probability (BPP) was >0.9, the results were considered credible. Consensus trees were viewed and edited using FigTree v.1.4.2 [57].

### 4.5. Divergence Time Estimates

Divergence times were estimated in BEAST v2.4.7.0 (Auckland, New Zealand) [58]. The best partitions and models were calculated in PartitionFinder v.2.1.1 [53]. Due to no fossils to calibrate, the inferred time was 134 MYA from Dietrich et al. [4]. A relaxed clock log model and the Birth–Death Model were employed for speciation events. The MCMC was run for 100 million generations; trees were sampled for every 1000 generations. Discarding the first 25% of the run, the maximum clade credibility tree with median heights coupled with 95% highest posterior density heights interval (95% HDP) was summarized in TreeAnnotator v.1.8. TRACER v.1.7 [56] was used to assess convergence and estimate the effective sample size (ESS) of each parameter.

### 4.6. Ancestral State Reconstructions

Ancestral state reconstructions based on the time tree generated in the last step using Bayesian Binary MCMC analysis (BBM) with RASP (Reconstruct Ancestral State in Phylogenies) v.4.0. (Sichuan, China) [59]. Pronotum characteristic was scored as a binary character with states: (A) pronotum with suprahumeral horn; (B) pronotum without suprahumeral horn; (C) others.

## Figures and Tables

**Figure 1 ijms-24-00694-f001:**
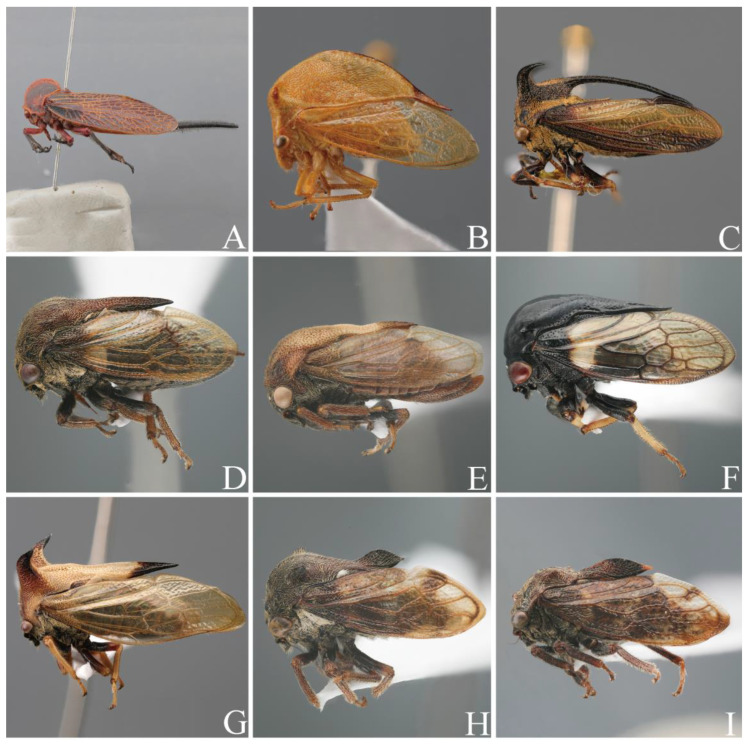
Specimens representative of Membracidae: (**A**) *Darthula hardwickii* (Aetalioninae: Darthylini); (**B**) *Stictocephala bisonia* (Smillinae: Ceresini); (**C**) *Leptocentrus albolineatus* (Centrotinae: Leptocentrini); (**D**–**I**) *Centrotoscelus davidi*, *Kotogargara minuta*, *Tricentrus fulgidus*, *Tricentrus gammamaculatus*, *Antialcidas floripennae*, and *Machaerotypus stigmosus* (Centrotinae: Gargarini).

**Figure 2 ijms-24-00694-f002:**
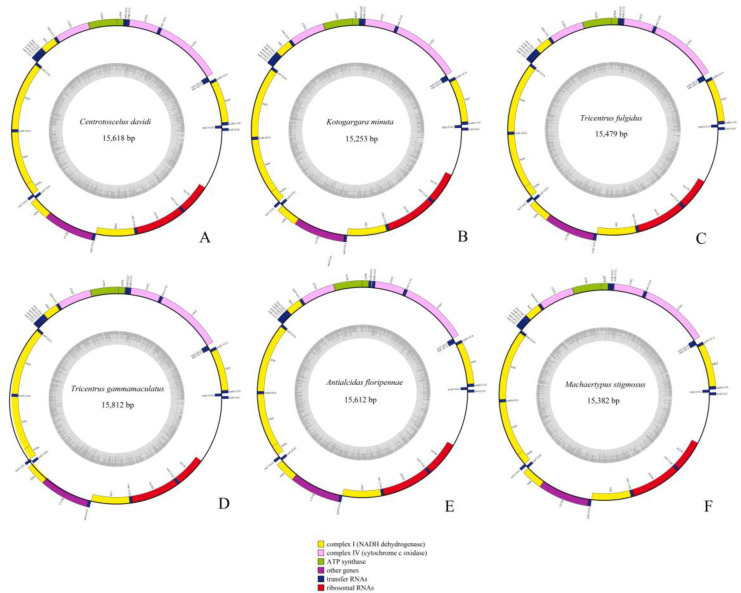
Circular maps of the mitogenomes of Gargarini. (**A**) *Centrotoscelus davidi*, (**B**) *Kotogargara minuta*, (**C**) *Tricentrus fulgidus*, (**D**) *Tricentrus gammamaculatus*, (**E**) *Antialcidas floripennae*, (**F**) *Machaerotypus stigmosus*. Color blocks outside the circle indicate that the genes are on the majority strand (J-strand); those within the circle indicate the genes are located on the minority strand (N-strand).

**Figure 3 ijms-24-00694-f003:**
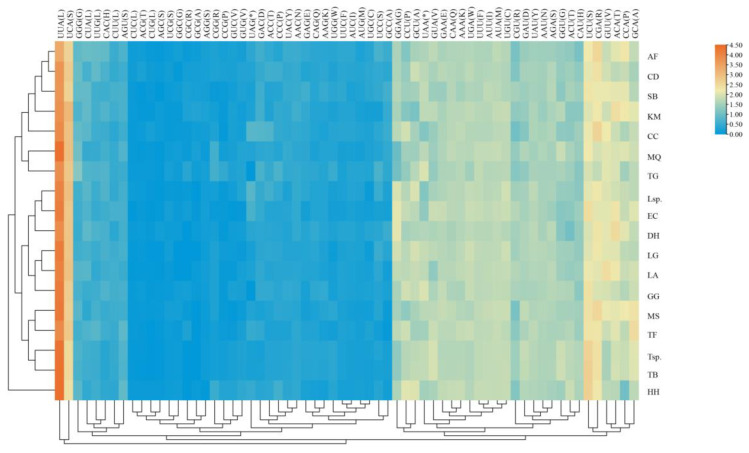
The relative synonymous codon usage (RSCU) of PCGs in the 18 treehoppers mitogenomes. The *x*-axis and *y*-axis indicate the hierarchical clustering of codon frequencies and treehoppers species, respectively. AF: *Antialcidas floripennae*, CD: *Centrotoscelus davidi*, SB: *Stictocephala bisonia*, KM: *Kotogargara minuta*, CC: *Centrotus cornutus*, MQ: *Maurya qinlingensis*, TG: *Tricentrus gammamaculatus*, Lsp.: *Leptobelus* sp. EC: *Entylia carinata*, DH: *Darthula hardwickii*, LG: *Leptobelus gazella*, LA: *Leptocentrus albolineatus* GG: *Gargara genistae*, MS: *Machaerotypus stigmosus*, TF: *Tricentrus fulgidus*, Tsp.: *Tricentrus* sp., TB: *Tricentrus brunneus*, HH: *Hypsauchenia hardwickii*.

**Figure 4 ijms-24-00694-f004:**
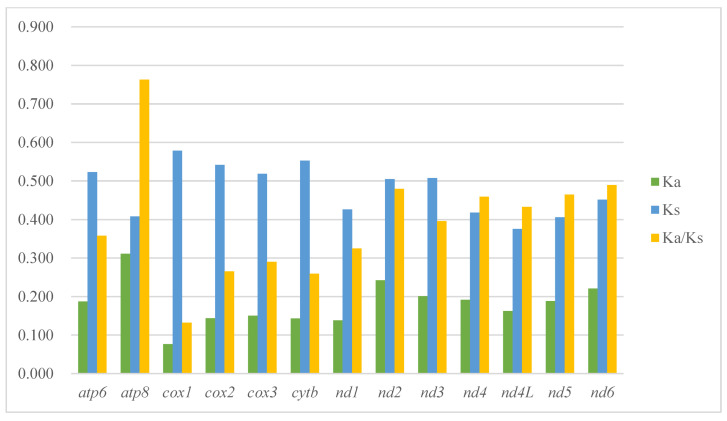
The evolutionary rate of each PCGs in the mitogenomes of 18 treehoppers in this study. Ka is the nonsynonymous nucleotide substitutions per nonsynonymous site, Ks is the synonymous nucleotide substitutions per synonymous site, and Ka/Ks is the ratio of the rate of nonsynonymous substitutions to synonymous substitutions.

**Figure 5 ijms-24-00694-f005:**
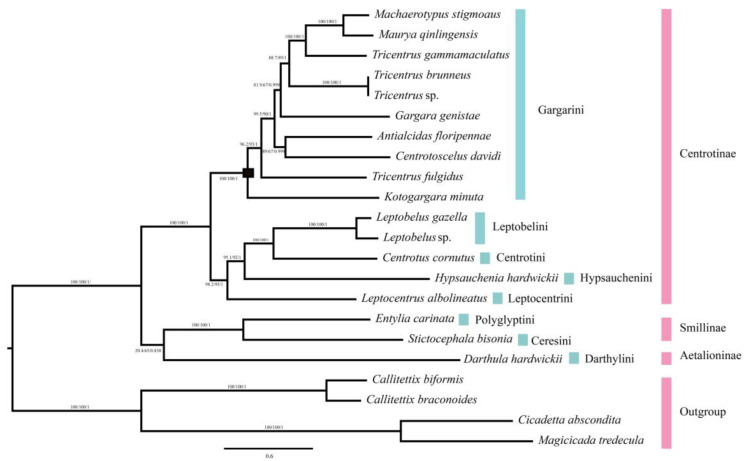
Phylogenetic trees of treehoppers were inferred using MrBayes (Bayesian inference) and maximum likelihood (ML) analysis based on the nucleotide sequences of the 13 protein-coding genes. Bayesian posterior probabilities (BPPs) and bootstrap percentages (BP) are indicated on branches. The black box indicates the branch of Gargarini tribe.

**Figure 6 ijms-24-00694-f006:**
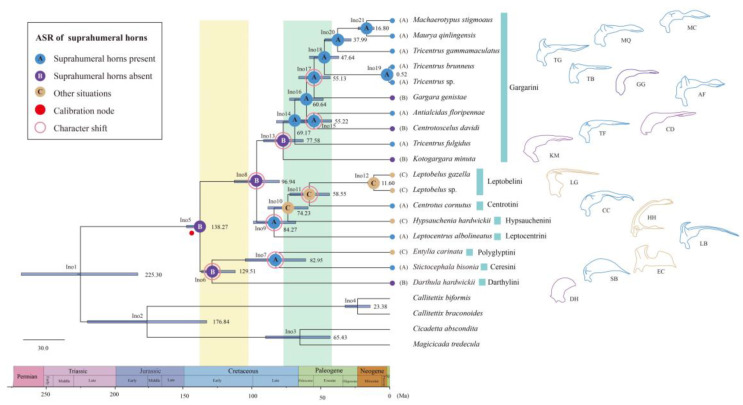
Chronogram showing the ancestral state reconstructions and temporal divergences of treehopper. Pie charts on nodes indicate most likely states only. Blue bars indicate time intervals for 95% probability of actual age (Appendix A). The color of nodes at the end of the tree and the color of letters in brackets correspond to states of treehopper suprahumeral horn on the left box of the figure. The figure on the right is the lateral view sketch of pronotum of the treehoppers. Calibration nodes by the red dot. Character shift by pink circles.

## Data Availability

Not applicable.

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
