# Peer review of "Revisiting the Phylogenetic Relationship and Evolution of Gargarini with Mitochondrial Genome (Hemiptera: Membracidae: Centrotinae)"

_ijms, 2022, doi:10.3390/ijms24010694_

Round 1

Reviewer 1 Report

In this manuscript, the authors try to investigate the phylogenetic relationship and evolution of Gargarini through comparision of mitochondrial genomes from various species, the results will provide some information for the researchs of evolution among insects. The MS was not prepared well and some issues should be concerned:

1.       In the abstract, the sentence“The length of mitochondrial genomes is 15253 bp to 16467 bp,” which one is 16467bp? This is not consistent with Figure 2.

2.       In the introduction, the Figure 1 could be listed as supplementary file. Instead, the figures of the six species used in this study should be added in Figure 2.

3.       The complete mitogenomes of six Gargarini species: Centrotoscelus davidi, Kotogargara minuta, Tricentrus fulgidus, Tricentrus gammamaculatus, Antialcidas floripennae , Machaerotypus stigmosus , and one Centrotini treehopper: Centrotus cornutus were identified as circular double-stranded molecules (Figure 2). So where is the Centrotus cornutus in Figure 2? Why Centrotini treehopper was included here, since the authors aimed to identify the Gargarini species?

4.       In 2.4. Control Region, ...‘ranging from 869 bp (C. cornutus) to 2261 bp (T. brunneus)’, where is T. Brunneus ??   ‘ ...with AT contents ranging from 72.7% (Tricentrus sp.) to 89.4% (G. genistae)’, where is the data?? this should be listed correspoding to each species.

5.       ‘It was located between s-rRNA and trnaI in 11 treehopper mitogenomes... ’‘the control region with structural organization of seven treehoppers mitogenomes is illustrated in Figure S8. Two repeat regions were present in C. davidi, T. gammamaculatus, T. fulgidus, and one in remaining four species, K. minuta, A. floripennae, M. stigmosus, and C. cornutus. So 11 or seven treehopper mitogenomes were used for comparison here? And results was not consistent with the Figure.

6.       The mitogenome sequences of these six species were seperately sequenced twice or only once? in Figure 5, all the mitogenome sequences from various species used for phylogenetic analysis are complete or incomplete? This is important for the conclusion.

7.       What the new findings in this MS? This should be focused in abstract. In addition, Supplementary Tables are not found. The format of references was inconsistent.

8.       The English should be furthe improved.

Reviewer 2 Report

I believe in open review - My name is Jeremy Wideman and I am an assistant professor at Arizona State University. I have expertise in mitochondrial genomes of protists and can provide some feedback, but am largely not qualified to review this paper. But what I can definitively say is that this paper doesn't fit the scope of IJMS and I don't know why an editor would send it for review. There are plenty of other journal's whose scope would fit better. This is the only reason I suggest rejection. What is the point of a journal scope if it isn't followed by the editors?

The mitochondrial genome figure annotations really need to be enlarged.

Beyond the mitochondrial genome assembly and stats, I am not qualified to review the other aspects of the paper as I am an evolutionary cell biologist and do not know anything about animals or insects.

Round 2

Reviewer 1 Report

I think the author don't answered the questions completely or correctly, for example, questions 2,3,6 8. and the MS could be further improved by native English speaker.

Reviewer 2 Report

Again, the paper is fine, but I don't think belongs under the scope of IJMS
